# Self-Injurious Behavior and Its Characteristics in a Sample of Mexican Adolescent Students

**DOI:** 10.3390/healthcare11121682

**Published:** 2023-06-07

**Authors:** Gilda Gómez-Peresmitré, Romana Silvia Platas-Acevedo, Rodrigo León-Hernández, Rebeca Guzmán-Saldaña

**Affiliations:** 1Faculty of Psychology, The National Autonomous University of Mexico, Av. Universidad 3004 Col Copilco-Universidad, Alcaldía, Coyoacán, Mexico City C.P. 04510, Mexico; 2National Council of Science and Technology, Avenida Insurgentes Sur 1582, Crédito Constructor, Ciudad de México C.P. 03940, Mexico; 3Institute of Health Sciences, Autonomous University of the State of Hidalgo, Camino a Tilcuautla s/n Pueblo San Juan Tilcuautla, Hidalgo C.P. 42160, Mexico

**Keywords:** adolescence, depression, self-injurious-behavior, methods, motives, time and frequency

## Abstract

Adolescence is a difficult stage, a period of risk for developing disorders, including depression and self-injurious behavior. A non-random sample was drawn (*n* = 563) from first-year high school students (32.8%) 185 males and 378 females (67.14%) from public schools in Mexico. The age range was 15 and 19 years, with a mean age of 15.63 (SD = 0.78). According to the results, the sample was divided into n1 = 414 (73.3%) adolescents without self-injury (S.I.) and n2 = 149 (26.4%) S.I. adolescents. In addition, results were obtained on methods, motives, time, and frequency of S.I., and a model was generated in which depression and first sexual experience obtained the highest Odd Ratio and d values in their relationship with S.I. Finally, we contrasted the results with earlier reports and concluded that depression is an important variable in S.I. behavior. Early S.I. detection will prevent the aggravation of S.I. and suicide attempts.

## 1. Introduction

Non-suicidal self-injury (NSI and NSSI) or “self-injury” refers to the direct and deliberate destruction of body tissue without any observable intent to die. Self-injury is all those injuries deliberately inflicted on one’s body without suicidal intent [1,2]. Although there is still no consensus to classify subtypes of self-injury, researchers and clinicians agree that the behavior usually ranges on a scale from mild (low frequency) to moderate (more frequent and severe, with possible medical requirement), and severe (high frequency, severe injury, and resulting impairment) [3,4].

Adolescence is a difficult stage of life. It represents a period of risk for the development of disorders such as depression and anxiety [5,6], both because of the biological, psychological, and social changes that increase intense emotional experiences and because of the social challenges and stressful life events that adolescents have to face if they are to achieve good adaptive functioning [7,8].

Self-injurious (S.I.) behaviors are direct behaviors (e.g., cutting or burning the skin, hitting body parts such as the head) and indirect behaviors (substance abuse, risky sexual behaviors, or eating disorders) [9]. In indirect behaviors, the primary motivation does not directly seek to feel pain or visibly damage the skin, and usually, the behavior and the physical consequence are not instantly and unequivocally related, as is often the case among direct behaviors [10,11].

Some research points to the goal of self-injury (S.I.) being variable, relieving negative emotions that the individual may suffer, such as depression, anger, and anxiety [12,13], regulating negative affect, reducing stress, expressing emotions symbolically, self-punishing, obtaining care and attention from others, stopping dissociative episodes, even resisting a suicide attempt [14,15,16]. In clinical populations, prevalence assessments are between 30% and 40% of adolescents requiring psychiatric hospitalization because they have provoked some S.I. behavior [17]. Some community research on the general population reports that this type of behavior is committed by 13.01–29.02% of adolescents at least once in their lifetime [18,19]. In Germany, Brunner et al. [20], a population-based study showed that 4% of adolescents who engage in S.I. behaviors frequently repeat them more than four times a year. Nock and Prinstein [16] and Whitlock and Knox (2007) [21] place the age of initiation between 10 and 15, although for other authors, it is between 12 and 17 years [22] and affirms that the onset of these behaviors after the age of 30 is infrequent. In 2015 in Spain, 32.20% of 1864 adolescents were reported to have engaged in severe S.I., and 55.60% were classified with S.I. In total, 48.01% of the sample accepted having, at some time, bitten themselves, although this behavior was perceived as acceptable and usual [23]. Klonsky et al. [24] found that it is usual for most self-injurers to use more than one method.

In Mexico, in an investigation on deliberate S.I. in a sample of schoolchildren, the most representative symptoms were detected as cutting the skin with sharp objects, burning the skin with a cigarette, inserting a needle, pin, staple, or another object into the skin, preventing a wound from healing or intoxicating oneself with poison, gas, or other substance [25]. Albores-Gallo et al. [26] conducted research in Mexico City (CDMX). They found a prevalence of S.I. of 6% in adolescents (average age of onset of 12 years), although it was detected that it could occur even at ages as young as six years. The prevalence increases to 17% when considering repetitive behaviors that adolescents do not consider as severe S.I. Mayer-Villa et al. [27] conducted a study in a school population in Mexico City and found that 28% of adolescents self-injure, noting that these are similar percentages to those in other countries [26,28,29].

Regarding the sex of adolescents who have S.I., there is a relative consensus for clinical and community samples that S.I. behaviors are more frequent in females than males [30,31,32]. However, some community research has found no differences between sexes [33,34], and others have only shown them with some of the methods used, with a predominance of self-injury by cutting in the case of females [35] or burns in males [36]. It is important to note, concerning the higher frequency of S.I. in adolescent females, that additional clinical and community research has found that gender differences can be partially explained by the more significant presence of depressive symptomatology, low self-esteem, and emotional dysregulation in the female sex [37,38]. One of the purposes of the research conducted in this field is to generate models explaining why adolescents self-injure. Hielscher et al. [12] generated a model based on emotional regulation that indicates: (1) adolescents that report higher levels of negative affect than those who do not S.I., (2) after experiencing negative affect, the adolescent proceeds with self-injurious behavior, (3) decreased negative affect and relief are present after S.I., and (4) most adolescents whom S.I. seek relief from negative affect, as well as continued difficulties regulating other emotions [12,39]. Self-mutilation can be defined as an attenuated form of suicide (“focal suicide”). Thus, self-mutilation of an anti-suicidal act allows patients to come out of their dissociation and feel alive again. Therefore, the risk of suicide will not increase because self-mutilation produces the expected relief. However, most authors exhibit self-mutilation as a risk factor for completed suicide. Self-mutilation behavior differs from suicide attempts by the lack of systematic suicidal intentions [40].

As seen, self-injury (S.I.) is an emotional regulation strategy in which impulsive acts provide supposed relief from intense negative emotions [41]. Although emotional dysregulation seems to be a risk factor that facilitates and maintains S.I., by itself it does not explain the reason for S.I. since there are adolescents who present negative affects and do not S.I., suggesting that other psychological processes should be sought to understand why people S.I. [42]. Some risk factors are altered body representations, as it has been related to other disorders such as eating disorders, body dysmorphia, and neuropathic pain [12,43]. According to Cash and Pruzinsky [44], altered body representations and sensory perceptions may be risk factors for adolescents who already self-injure, i.e., when the body becomes more susceptible to the self-concept (generating body dissatisfaction by promoting the body as a hated object, making it easier to harm) [45,46,47]. This research aims to know the prevalence of self-injurious behavior in adolescent male and female first-year high school students. It is postulated that between the variables of self-injury and anxiety, depression, emotional dysregulation, and risk eating behavior (REB), there are statistically significant relationships when the first is considered as the dependent variable (DV) and the others as independent variables (IVs). As part of this purpose, we also want to know which of the I.V.s has the greatest weight or influence in its relationship with the S.I. variable, and finally, we want to find the best model that explains this relationship.

## 2. Materials and Methods

### 2.1. Sample

Design: The present study corresponds to a correlational, retrospective design, with non-probabilistic sample of *n* = 563 first year high school students, 185 males (32.8%) and 378 females (67.14%) from public schools in Mexico. The age range was between 15 and 19 years, mean age 15.63 (SD = 0.78). According to the results, the sample was subdivided into n1 = 414 (73.3%) adolescents without self-injury and n2 = 149 (26.4%) self-injured adolescents.

### 2.2. Instrument

The scales of the online test for the self-assessment of risk factors for eating disorders, depression, social anxiety, and self-injury [OTESSED; Ref. [48]] were used.

A sociodemographic section, consisting of 20 items that explore data such as age, work, parents’ level of education, parents’ occupation, sexual orientation, sexual life, and first sexual experience.

Depression and Suicidal Ideation Scale Section. Consists of 11 multiple choice questions ranging from “Never” to “Always”, e.g., I feel that it is not worth continuing in this world, When I am sad, I think about death. It has a coefficient of ϖ = 0.86 for men and ϖ = 0.89 for women.

Risk Factors Associated with Eating Disorders Scale (EFRATA) by its Spanish acronym Section. It has 11 items with five response options ranging from “never” = 1 to “always” = 5 (higher score implies greater problem or risk) that explore compulsive eating behavior, normal eating behavior, chronic dieting, and feelings of guilt. It has a coefficient ϖ = 0.88 for men and ϖ = 0.84 for females.

### 2.3. Social Anxiety Scale Section

It has 12 items with multiple-choice response forms, ranging from “never” to “Always”. It has a coefficient ϖ = 0.88 for males and ϖ = 0.87 for females.

Section K. Self-Injury Scale. This section has 27 statements measuring the frequency, amount, and timing of S.I. Have you ever deliberately self-injured or intentionally cut your skin? How many times have you intentionally cut your skin?

### 2.4. Preoccupation with Body Weight Section

It has 14 items that measure concern about weight, e.g., Do you worry about your body weight? At what age did you start to worry about your body weight?

#### Procedure

Given the worldwide health contingency due to the SARS-CoV-2 virus, which led to confinement, the instrument was applied online. The directors of different high school campuses were asked for their student’s participation. Students who decided to take part were sent the link to the instrument via e-mail. This study adhered to the Helsinki Declaration, to the General Health Law on Research that governs our country, with the consideration that it is a risk-free research, and was approved by the ethics committee of the Faculty of Psychology (FPCE_ 08032021_H_AC 27/04/2021). The International Test Commission (ITC) guidelines for online evaluations were followed, as well as those required for UNAM institutional website.

### 2.5. Statistical Analysis

The Statistical Package for the Social Sciences [SPSS], V. 22 (IBM, MEXICO) was used. Pearson’s correlation analysis was performed to establish the relationship between variables: Risk Eating Behavior (REB), Depression, Anxiety, Body Weight Concern (BWC), Emotional Dysregulation, Sexual Orientation, Sexual Life, and First Sexual Experience. Chi-square was used to detect statistically significant differences between the variable sex, methods of S.I., reasons for S.I., and causes and time of S.I. A binary logistic regression analysis was also performed to evaluate the relationship between the predictor variables of self-injury.

## 3. Results

It is important to find the relationship between the sociodemographic variables with the variables under study because the sociodemographic variables constitute the influence of the environment or background in which the participant’s daily life develops. Age, family relationship, work, and sex life are some of these variables (See Table 1).

### 3.1. Self-Injury Methods

Cross-table analyses were performed to determine the frequency of the most commonly used S.I. method. Among the most common were, for women, scratching the skin 89% (*n* = 75), injuries with suicidal thoughts 87. 4% (*n* = 83), and impeded healing 88% (*n* = 66). For the male sample the most frequent S.I. were cut 17. 8% (*n* = 18) and hit of the head 22.4% (*n* = 15), lastly, considered within the S.I. method were injuries with suicidal thoughts 12.6%(*n* = 12 among males vs. 87.4 (83) in females. Four S.I. methods showed significant differences (*p* ≤ 0.05), pinching to the body, scratching, hair swallowing, and injuries with suicidal thought (See Table 2).

### 3.2. Motives for Self-Injure

Among the reasons reported by adolescents for S.I. are, among females, to try to feel better 94.2% (*n* = 65), to take away bad memories or sad things (86.2%) (*n* = 56), and to not think about problems 91.4% (*n* = 53). The group of men indicated that the main causes were to take away bad memories or sad things 13.8% (*n* = 9), out of curiosity 16.7% (*n* = 7), and to get attention 38.5% (*n* = 5). The causes that showed statistically significant differences (*p* ≤ 0.05) by sex were to try to feel better, to not think about my problems, to punish myself, and to get attention (See Table 3).

### 3.3. Time since Last Self Injury

Females reported the highest percentage of the response “more than one year ago”. The methods mentioned were cutting 36.3% (49) and biting 26.9% (36). For the option yesterday–last week the highest percentage was nail biting 14.8 (*n* = 20), body pinching 12.6% (*n* = 17), and S.I. with suicidal thoughts. For the group of men, for the response more than a year ago, the highest percentage was cutting with 57% (*n* = 16, thoughts with suicidal intent in 35% (10), and skin tagging 32.1% (9). Statistically significant differences were found between the time and the methods of SI: head banging, bruising, biting, and S.I. with suicidal thoughts (*p* ≤ 0.05) (See Table 4).

### 3.4. Frequency of Self Injury

In the responses given for the frequency of the last S.I., females reported the highest percentage for the response more than three times and for the behavior cutting 40% (54), pinching the body 29.6% (40), and scratching the skin 30.4% (41). For the male group, the highest percentage in the frequency of more than three times was for cutting with 25% (7) and thoughts with non-suicidal intent with 17.9% (5). Statistically significant differences were found between the frequency of S.I. and methods of S.I. (bruising, preventing scarring, and scratching the skin (*p* ≤ 0.05) (See Table 5).

### 3.5. Logistic Regression

In order to confirm previous results, as well as to determine the risk predictors of self-injurious behavior, a binary logistic regression analysis was performed to evaluate the weight or effect of the variables studied as predictors of self-injury. The dependent variable was the presence or absence of self-injury, and the independent or predictor variables were risky eating behavior (REB), body weigh concern (BWC), anxiety, depression, dysregulation, sexual orientation, sexual life, and first sexual experience.

As can be seen in Table 6, the variables depression, REB, BWC, sexual orientation, sexual life, and first sexual experience interact to predict S.I. behavior, although with different effects. The variable S.I. presented a large effect in its relationship with depression; It is 8.79 times (OR [95% CI] = 8.79 [2.6–29.4] more likely for S.I. behavior to occur with the change of one unit of the variable depression; if the association is considered as Magnitude of Effect (M.E.), i.e., as a function of a d transformation the obtained value of OR indicates that it is a large effect. The OR value of the rest of the variables shows, in each case, a weak effect, such as that of REB; 2.19 times (OR [95% CI] = 2.19 [1.3–3.5]; that of BWC; 1.4 times (OR[95% CI] = 1.4 [1.2–1.7] more likely to have a sexual orientation other than heterosexual; and 1.8 times more likely to be sexually active (OR[95% CI] = 1.8 [1.2–2.9] as well as 0.3 times more likely to have had a first sexual experience in a nonconsensual manner (OR[95%CI] = 0.31 [0.10–0.97]. In Table 7, the standard deviation and average are presented by groups (S.I. vs. NSI).

A Kruskal–Wallis test was run to determine whether there is a statistically significant difference between the causes of self-injury and the variables in the regression model. The results indicated that the significant variables were Depression, REB, and sexual life (*p* ≤ 0.05) (See Table 8).

## 4. Discussion

Knowing the prevalence of Self-Injurious (S.I.) behavior in Mexican adolescent high school students, as well as the relationship between the independent variables and S.I., the dependent variable, and determining the weight of each of them (finding a model) in that relationship, were some of the research purposes. The prevalence of the S.I. variable for women resulted in twenty-four percent, a value close to the twenty-eight percent reported by Mayer et al. [27] as a result of a study also conducted in the Mexican population. It is important to note that such a percentage is similar to that of other countries, as stated by Plener et al. [28], Yates et al. [29], and Albores-Gallo et al. [26]. In contrast, the prevalence of S.I. in men was very low, at five percent. It should be noted that there is not much information in this regard. More studies are needed to know the relationship (men—SI).

The sample consisted of male and female adolescent students (fifteen years old on average), with more female self-injurers, coinciding with the results of other studies such as those of Morey et al. [31], O’Connor et al. [32], and Landstedt and Gillander, [30]. With respect to sexual orientation, more bisexual boys/girls were found among S.I. than on NSI. In this regard, it can be said that there is little research carried out on the sexual life/S.I. of students in this stage of life, adolescence. Taking into account heterosexual orientation, more boys and girls were found and more than half of the participants without an active sexual life. Among the NSI, there are more students with first sexual experience between the ages of twelve and fifteen years or older. NSI students are more concerned about their body weight than S.I. More than half of NSI children live with their immediate family, a result consistent with other studies that show that there are more S.I. among children who live with only one parent [49]. Likewise, it was found that the more education the father had and the more he worked in formal economic activity, the more NSI were found.

Importantly, the results of this study are consistent with previous research [50,51,52,53,54,55] that has examined the relationship between parental economic status and the risk of self-injury in adolescents. Numerous population-based studies have identified socioeconomic disparities in the frequency of suicide [56,57] and self-harm [58,59,60]. In general, these studies have evidenced higher rates of suicidal behavior in groups with lower socioeconomic status compared to those with higher economic status in key aspects such as income, educational attainment, and employment [61]. These discrepancies also persist throughout life [61]. However, a universal presence of socioeconomic differences has not been found [62,63]; likewise, it has been found that the associations between economic status and suicide are less significant in women [61] or may even follow an opposite direction (higher incidence of suicide in women with a higher economic status) [64,65]. Several studies point out that lower education on the part of the mother is associated with a higher risk of suicidal behavior in individuals [59,60]; however, other studies have found higher maternal education [61,64] and educational attainment [65] associated with a higher risk of suicidal behavior. A possible explanation for the higher risk of unintentional self-injury associated with higher maternal educational attainment (and parental educational attainment in general) may be related to lower self-esteem and stress in those who do not meet the higher demands set by more highly educated parents [50,63].

Likewise, these findings may support the hypothesis that risk behavior is part of a general developmental process of adolescence associated with other behaviors such as poor management of emotions and is not related to socioeconomic status but rather to emotional and gender issues. More research is needed to prevent self-injury where adolescents and the socioeconomic status of parents should be considered as a possible target group.

Practically the same proportion of domestic violence was reported between S.I. and NSI, although more than half of the NSI students responded that they did not face domestic violence.

The survey also asked about eighteen procedures or methods of S.I., as well as their frequency. It can be seen in Table 2 that each item in the column dedicated to the female sex stands out for its higher percentages compared to those of men. This higher female response has been explained by a greater presence of depressive symptomatology [37,38]. This theoretical proposal is consistent with the data from the present study. It can be seen in Table 6 of correlations that depression (together with anxiety) shows the highest and most significant correlation with S.I. The higher value can also be seen in the logistic regression results (Table 7). It should be noted that only in four S.I. methods statistically significant differences were found (pinching, scratching, hair swallowing, and suicidal thinking), with female scores being the highest. Although it can be said that, in bruising and preventing healing, a trend is shown (values very close to statistical significance) [66] since these types of scores that differ by hundredths are usually interpreted as a trend.

While it is true that there is no consensus or knowledge regarding the possible causes that lead a boy/girl to S.I., the theory and research conducted [15,16,39,67,68] as well as the coincident results of this study, point out that most frequently reference is made to the liberating and regulating role of the feeling of negative affect. The feeling of relief produced, even if temporary, reinforces its repetition. Likewise, the facilitation of the expression of emotions, such as self-punishment and the search for means (what to do) to attract the attention of others, covers the purposes of relief so sought by those who S.I. It is important to note that these causes were notoriously and mostly accepted by women, although these marked differences were not so marked in relation to the cause or purpose of attracting attention. Attention seeking (also called distress) among men increased and reached the highest percentage among male causes of S.I.

More men than women were found to have more than one year of having self-hit their head, and in that same proportion have done so within the last week. Conversely, more women than men have self-injured in the last year and during the last week, while more men than women have done so recently. It is important to note that there appear to be almost gender-exclusive, i.e., exclusively female or male methods and causes. For example, the bruising method is applied by women, this is a method used exclusively by women, and the bruises can be the result of pinching or hitting. Women pinch (it seems to be a purely feminine action); mothers usually pinch their children as a sign of anger or punishment, but not fathers. If a man pinched, “it would look very feminine”. Women pinch themselves in practically the same proportion, whether in the past (a year or more) or in the present (last week). Could it be said that among women, this activity is perceived as mild, everyday, unimportant? According to Calvete et al. [23] and Klonsky et al. [24], this behavior is perceived as acceptable and usual by most participants in their studies. Introducing objects under the skin is an activity equally performed by men and women, with low presence both in the past (a year ago or more) and in the present (last week). In addition, few students chose this option as a response. Finally, there is the behavior of S.I. with suicidal ideation. Suicidal ideation is the most worrisome behavior. Women were found to have suicidal ideation to a greater extent compared to men. This finding should alert everyone related to health to find out what causes or reasons are behind this behavior. Why do women think more about suicide, although men are the ones who practice it to a greater extent? Although the answer is unknown, suicidal thinking has effects on other areas of the brain and on behavior that must be known for prevention.

It is important to note that the highest percentage found among women stands out above men. In this regard, there are two proposals related to the role of suicidal ideation in the consummation of suicide: the risk of suicide does not increase to the extent that S.I. produces the expected effect (e.g., regulating negative affect, reducing stress, obtaining care and attention) or suicidal ideation is a risk factor for the consummation of suicide [15,16,51]. More research is needed on these propositions that revolve around the gestation, maintenance, and prevention of S.I.

Among women, a higher frequency of bruising was found (more than three times) than among men. Another apparently all-female reported behavior was preventing scarring. In skin scratching, more women reported it more frequently, and more men reported doing it less frequently.

When analyzing the associations between independent variables and S.I., the DV (dependent variable), the strongest relationship was (depression →S.I.) confirming, as already noted, that the stronger weight of depression is above the role of another emotion, at least of those reported in most of the studies conducted. It can be seen (Table 5) that it was followed in importance by the variable first sexual experience; anxiety and risky eating behavior were also shown to be important in this relationship. It should be noted that although the values of the correlations can be evaluated as mean values, the magnitude of these values is considered acceptable when dealing with more complex sciences that are not hard sciences [68]. According to St. Germain and Hooley [9], risky eating behavior (REB) also correlates significantly with S.I. It is considered one of the indirect behaviors that is distinguished from direct behaviors by not seeking to cause immediate physical harm, as noted by Claes and Muehlenkamp and Claes and Vandereycken [10,11].

Analyzing the model that best explained the relationship between the Independent Variables (depression, REB, BWC, sexual orientation, sexual life, and first sexual experience) and the Dependent Variable (S.I.), through the Logistic Regression the previous results were confirmed, in that the depression variable was the one that showed the greatest weight in the DV much higher than that of the rest of the IV. It was followed in importance by a relationship with the variable, first sexual experience. Although in the last four variables (concern about body weight, risky eating behavior, sexual life, and sexual orientation) the OR values were low, it should be noted that they were also significant.

## 5. Conclusions

Among the most important results, we found, in the sample investigated, that among Mexican adolescent students the prevalence of S.I. resulted was very similar to the results reported in other studies conducted in Mexico and other countries. It was found that self-injurious behavior is primarily a female response and that certain methods of S.I. are exclusive to women, such as, cutting, suicidal thinking injuries, and scratching the skin. The detection of Self Injury behavior is difficult because of the social stigma. In this population, the main causes among male and female were trying to feel better, to punish myself, and to get attention.

In the association between the predictor variables and S.I., the founded model showed that depression maintains the strongest relationship, followed by the first sexual experience, body weight concern, risky eating behavior, and sexual orientation. In addition, obtaining the model that best explains the relationship between the predictor variables and S.I. behavior allowed us to prove the importance of the depression variable by showing the strongest effect (OR and d), which is a significant increase in S.I. for each unit of change in the depression variable. This model helps the detection of S.I. behavior in adolescents. Early detection of S.I. will facilitate the development of preventive and intervention plans in order to prevent suicidal ideation, suicide attempts, and completed suicide. Designing assistance, prevention, and guidance programs in school sectors is an important aspect in the prevention of these behaviors among adolescents.

### Limitations and Advances

The main limitation is the non-random nature of the sample of the students, which prevents the generalization of the results to the population from which the sample was drawn. The self-response format is also a limitation. On the other hand, the results allowed us to detect the importance of resuming and expanding the study of certain variables, such as those of a sexual nature, taking into account the vulnerability of this population in this area and at this stage of life.

## Figures and Tables

**Table 1 healthcare-11-01682-t001:** Description of Sociodemographic Variables by S.I. Group.

Variable	Self Injury		No Self-Injury	
	Men	Women	Men	Women
Range age	15–19	15–19	15–19	15–18
Average (D.E.)	15.81 * (1.22 **)	15.72 * (0.72 **)	15.59 * (0.79 **)	15.69 * (0.68 **)
BMI (WHO cut-off points)	2.31 * (0.77 **)	2.21 * (0.67 **)	2.13 * (0.69 **)	2.11 * (0.62 **)
Low weight	3 (13.6%)	12 (9.4%)	25 (15.3%)	27 (10.8%)
Normal weight	10 (45.5%)	82 (64.6)	96 (58.9%	177 (70.5%)
Overweight	8 (36.4%)	37 (21.3%)	37 (22.7%)	38 (15.1%)
Obesity	1 (4.5%)	9 (3.6%)	5 (3.1%)	9 (3.6%)
Sexual orientation	1.59 * (0.90 **)	1.76 * (0.87 **)	1.31 * (0.70 **)	1.24 * (0.56 **)
Heterosexual	14 (63.5%)	56 (44.1%)	132 (81%)	201 (80.1%)
Bisexual	4 (18%)	56 (44.1%)	14 (8.6%)	43 (17.1%)
Homosexual	3 (13.6%)	4 (3.1%)	14 (8.6%)	3 (1.2%)
Another	1 (4%)	11 (8.7%)	3 (1.8%)	4 (1.6%)
Sexual active life	1.36 * (0.79 **)	1.39 * (0.79 **)	1.24 * (0.65 **)	1.24 * (0.65 **)
Yes	4 (18.2%)	25 (19.7%)	20 (12.3%)	28 (11.2%)
No	18 (81.8%)	102 (80.3%)	143 (87.7%)	223 (88.8%)
Age of first sexual experience	0.63 * (1.39 **)	0.25 * (0.92 **)	0.44 * (1.21 **)	0.24 * (0.89 **)
Less than 9 years old	-	1 (0.8%)	-	-
9–11 years old	-	-	1 (0.6%)	1 (0.4%)
12–14 years old	2 (9.1%)	2 (1.6%)	5 (3.1%)	9 (3.6%)
15 years or more	2 (9.1%)	6 (4.7%)	14 (8.6%)	8 (3.2%)
Not specified	-	16 (12.6%)	-	10 (4%)
Have carried out restricted diet	1.5 * (0.91 **)	1.72 * (0.96 **)	1.25 * (0.67 **)	1.33 * (0.74 **)
Yes	6 (27.3%)	46 (36.2%)	21 (12.9%)	42 (16.7%)
No	16 (72.7%)	81 (63.8%)	142 (87.1%)	209 (85.3%)
They care about their weight	2.4 * (0.91 **)	2.77 * (0.62 **)	2.34 * (0.94 **)	2.6 * (0.80 **)
Yes	16 (72.7%)	113 (89%)	109 (67.3%)	201 (80.1%)
No	6 (27.3%)	14 (11%)	53 (32.7%)	50 (19.9%)
Currently living with	1.3 * (0.49 **)	1.3 * (0.5 **)	1.25 * (0.46 **)	1.20 * (0.44 **)
Nuclear Family	14 (63.6%)	88 (69.3%)	124 (76.1%)	204 (81.3%)
Father or Mother	8 (36.4%)	37 (29.1%)	37 (22.7%)	43 (17.1%)
Another	-	2 (1.6%)	2 (1.2%)	4 (1.6%)
Father’s education	3.45 * (1.14 **)	3.37 * (0.98 **)	3.34 * (1.00 **)	3.11 * (0.99 **)
I do not have a father	6 (27.3%)	15 (11.8%)	24 (14.7%)	27 (10.8%)
Uneducated	1 (4.5%)	-	-	2 (0.8%)
Basic education	2 (9.1%)	31 (24.4%)	40 (24.5%)	78 (31.1%)
Middle education	11 (50%)	32 (25.2%)	51 (31.3%)	87 (34.7%)
Higher education	2 (9.1%)	49 (38.6%)	48 (29.4%)	57 (22.7%)
Father’s occupation	2.77 * (0.92 **)	2.36 * (0.70 **)	2.49 * (0.78 **)	3.10 * (0.98 **)
I do not have a father	7 (31.8%)	14 (11%)	30 (18.4%)	26 (10.4%)
Unemployed	-	3 (2.4%)	-	9 (3.6%)
Formal economic activity	12 (54.5%)	89 (70.1%)	113 (68.3%)	174 (69.3%)
Informal economic activity	3 (13.6%)	21 (16.5%)	20 (12.3%)	42 (16.7%)
Mother’s education	2.86 * (0.88 **)	3.18 * (0.74 **)	3.13 * (0.82 **)	2.9 * (0.84 **)
I do not have a mother	-	-	6 (3.7%)	3 (1.2%)
Uneducated	-	4 (3.1%)	-	4 (1.6%)
Basic education	10 (45.5%)	24 (18.9%)	39 (23.9%)	76 (30.3%)
Middle education	5 (22.7%)	52 (40.9%)	69 (42.3%)	93 (37.1%)
Higher education	7 (31.8%)	47 (37%)	49 (30.1%)	75 (30%)
Mother’s occupation	1.54 * (0.50 **)	3.11 * (0.82 **)	1.53 * (0.56 **)	2.9 * (0.84 **)
I do not have a mother	-	-	6 (3.7%)	3 (1.2%)
Unemployed	-	-	-	--
Formal economic activity	12 (54.5%)	73 (57.5%)	76 (46.6%)	137 (54.6%)
Informal economic activity	10 (45.5%)	54 (42.5%)	81 (49.7%)	111 (44.2%)
Domestic violence	2.18 * (1.0 **)	2.0 * (1.0 **)	2.75 * (0.65 **)	2.43 * (0.90 **)
Yes	9 (40.9%)	58 (45.7%)	20 (12.3%)	71 (28.3%)
No	13 (59.1%)	69 (54.3%)	143 (87.7%)	180 (70.7%)
Type of violence	1.66 * (1.0 **)	5.35 * (2.45 **)	2.4 * (1.0 **)	3.35 * (2.45 **)
Physical	5 (55.6%)	11 (/19%)	3 (15%)	8 (11.3%)
Psychological	3 (33.3%)	23 (39.7)	11 (6.7%)	11 (15.5%)
Economic	-	3 (5.2%)	1 (0.6%)	2 (2.8%)
Verbal	1 (11.1%)	8 (13.8%	5 (3.1%)	2 (2.8%)
All of the above	-	1 (1.7%)	-	-
Not specified	-	12 (20.7%)	-	48 (67.6%)

Note: * Average, ** SD (Standard Deviation).

**Table 2 healthcare-11-01682-t002:** Sex differences by methods of self-injury.

S.I. Procedure	Male % (n)	Female % (n)	Total	Chi2 o Fishers Exact Test	*p*
Cut	17.8 (18)	82.2 (83)	101	0.07	0.47
Burned	13.9 (5)	86.1 (31)	36	0.37	0.37
Bitten	17.2 (10)	82.8 (48)	58	0.00	0.57
Hit head	22.4 (15)	77.6 (52)	67	2.17	0.10
Bruised	11.8 (8)	88.2 (60)	68	2.4	0.08
Impede healing	12 (9)	88 (66)	75	2.6	0.07
Pinched IF body	31.3 (10)	68.8 (22)	32	5.5	**0.02**
Scratched skin	10.7 (9)	89.3 (75)	84	5.1	**0.02**
Pulled out hair	12.8 (6)	87.2 (41)	47	0.9	0.23
Broken bones	14.3 (1)	85.7 (6)	7	0.04	0.65
Swallowing hair	-	100 (20)	20	4.7	**0.01**
Swallowing something inedible	11.8 (2)	88.2 (15)	17	0.39	0.41
Hospitalization for SI	-	100 (9)	9	1.9	0.17
Suicidal thinking injuries	12.6 (12)	87.4 (83)	95	3.3	**0.05**
Nail biting	11.8 (6)	88.2 (45)	51	1.5	0.15
Spillage of something toxic	16.7 (2)	83.3 (10)	12	0.002	0.66
Marking on the skin	14.8 (9)	85.2 (52)	61	0.40	0.34
Inserting objects under the skin	9.1 (3)	90.9 (30)	33	1.9	0.12

Note: In bold, statistically significant data (*p* ≤ 0.05).

**Table 3 healthcare-11-01682-t003:** Sex differences by cause of self-injury.

Causes of SI	Male % (n)	Female % (n)	Total	Chi2 o Fisher’s Exact Test	*p*
Out of curiosity	16.7 (7)	83.3 (35)	42	0.01	0.56
To try to feel better	5.8 (4)	94.2 (65)	69	10.89	**0.001**
To get my mind off my problems	8.6 (5)	91.4 (53)	58	4.6	**0.02**
To stop feeling or diminish my feelings of loneliness	12.5 (5)	87.5 (35)	40	0.81	0.26
To punish myself	8.7 (4)	91.3 (42)	46	3.2	**0.05**
To show others how angry I feel	33.3 (2)	66.7 (4)	6	1.14	0.27
To push away bad memories or sad things	13.8 (9)	86.2 (56)	65	0.84	0.24
To hurt important people in my life	-	100 (4)	4	0.85	0.46
To be different from others	-	100 (2)	2	0.42	0.68
To get attention	38.5 (5)	61.5 (8)	13	4.5	**0.05**
To show others how bad I feel	20 (1)	80 (4)	5	0.02	0.61

Note: In bold, statistically significant data (*p* ≤ 0.05).

**Table 4 healthcare-11-01682-t004:** Self-injury by sex and time of self-injury.

S.I. Methods	Last Self-Injury Prevalence								Total	Chi2 o Fisher’s Exact Test	*p*
	More than One Year %(n)		6 Months to One Year %(n)		Last Month to More Than Three Times %(n)		Yesterday—Last Week %(n)				
	M	F	M	F	M	F	M	F			
**Cut**	**57.1 (16)**	**36.3 (49)**	3.6 (1)	12.6 (17)	3.6 (1)	10.4 (14)	-	3.7 (5)	103	6.44	0.16
Burned	14.3 (4)	13.4 (18)	-	6.0 (8)	3.6 (1)	3.0 (4)	-	0.7 (1)	36	2.00	0.73
Bitten	10.7 (3)	26.9 (36)	14.3 (4)	0.7 (1)	7.1 (2)	1.5 (2)	3.6 (1)	7.5 (10)	59	19.7	**0.001**
Hit head	28.6 (8)	7.4 (10)	3.6 (1)	13.3 (18)	3.6 (1)	13.3 (18)	14.3 (4)	4.4 (6)	66	17.37	**0.002**
Bruised	-	8.9 (12)	-	14.8 (20)	-	16.3 (22)	-	9.6 (13)	67	23.59	**0.00**
Impede healing	14.3 (4)	14.2 (19)	3.6 (1)	10.4 (14)	10.7 (3)	13.4 (18)	3.6 (1)	11.2 (15)	75	9.94	0.41
Pinched IF body	10.7 (3)	11.9 (16)	14.3 (4)	9.6 (13)	3.6 (1)	13.3 (18)	7.1 (2)	12.6 (17)	74	3.57	0.46
Scratched skin	10.7 (3)	12.6 (17)	7.1 (2)	12.6 (17)	10.7 (3)	20.7 (28)	3.6 (1)	10.4 (14)	85	5.87	0.20
Pulled out hair	3.6 (1)	11.2 (15)	3.6 (1)	3.7 (5)	3.6 (1)	12.7 (17)	10.7 (3)	11.9 (16)	59	4.32	0.36
Broken bones	-	4.4 (6)	-	-	-	-	-	-	6	1.23	0.32
Swallowing hair	-	5.9 (8)	-	4.4 (6)	-	1.5 (2)	-	3 (4)	20	4.72	0.31
Swallowing something inedible	3.6 (1)	9.6 (13)	-	1.5 (2)	-	0.7 (1)	3.6 (1)	-	18	6.49	0.16
Hospitalization for SI	-	3 (4)	-	3 (4)	-	0.7 (1)	-	-	9	1.91	0.57
Suicidal thinking injuries	35.7 (10)	20 (27)	3.6 (1)	17 (23)	-	11.9 (16)	3.6 (1)	12.6 (17)	95	12.33	**0.01**
Nail biting	7.1 (2)	5.9 (8)	10.7 (3)	3.7 (5)	3.6 (1)	8.1 (11)	-	14.8 (20)	50	7.60	0.10
Spillage of something toxic	7.1 (2)	6 (8)	-	-	-	0.7 (1)	-	-	11	0.26	0.87
Marking on the skin	32.1 (9)	17.8 (24)	-	8.1 (11)	-	11.1 (15)	-	3 (4)	63	1.24	0.06
Inserting objects under the skin	3.7 (1)	4 (5)	-	14.3 (18)	11.1 (3)	1.6 (2)	-	1.6 (2)	31	10.58	**0.03**

Note: In bold, statistically significant data (*p* ≤ 0.05).

**Table 5 healthcare-11-01682-t005:** Differences in Frequency of S.I. by gender.

S.I. Methods	Frequency S.I.		TOTAL	*p*
	More than Three Times %(n)			
	M	F		
Cut	25 (7)	40 (54)	102	0.23
Burned		2.2 (3)	37	0.64
Bitten	17.9 (5)	11.1 (15)	60	0.47
Hit head	17.5 (5)	17.6 (22)	56	0.12
Bruised	10.7 (3)	27.4 (37)	75	**0.04**
Impede healing		17.8 (24)	47	**0.001**
Pinched IF body	10.7 (3)	29.6 (40)	74	0.11
Scratched skin		30.4 (41)	85	**0.003**
Pulled out hair	14.3 (4)	22.2 (30)	59	0.18
Broken bones			7	0.65
Swallowing hair		5.2 (7)	20	0.09
Swallowing something inedible	3.6 (1)	3.7 (5)	18	0.69
Hospitalization for SI		1.5 (2)	9	0.37
Suicidal thinking injuries	17.9 (5)	33.8 (45)	93	0.15
Nail biting	7.1 (2)	17.8 (24)	50	0.35
Spillage of something toxic			12	0.66
Marking on the skin	7.1 (2)	14.8 (20)	63	0.53
Inserting objects under the skin	7.4 (2)	12.3 (16)	36	0.54

Note: In bold, statistically significant data (*p* ≤ 0.05).

**Table 6 healthcare-11-01682-t006:** Significant OR values (95% CI). Relationship between self-injury and independent variables.

		B	Wald	Sig.	Exp(B)	95% CI for EXP(B)	
						Lower	Upper
Step 1	Depression	2.17	12.46	0.000	8.79	2.63	29.42
	REB	0.69	5.72	0.017	2.01	1.13	3.56
	BWC	0.78	9.99	0.002	2.19	1.34	3.55
	Sexual Orientation	0.39	28.14	0.000	1.48	1.28	1.71
	Sexual Life	0.63	7.55	0.006	1.89	1.20	2.98
	First sexual experience	−1.16	4.03	0.045	0.31	0.10	0.97
	Constant	−10.56	45.11	0.000	0.00		

Note: Risk Eating Behavior (REB); Body Weight Concern (BWC).

**Table 7 healthcare-11-01682-t007:** Average and Standard Deviation for groups S.I. and independent variables.

	Self-Injury M (SD)		Not Self-Injury M (SD)	
	M	F	M	F
Depression	2.00 (0.01)	1.97 (0.15)	1.64 (0.48)	1.76 (0.48)
REB	1.81 (0.39)	1.86 (0.34)	1.60 (0.48)	1.63 (0.48)
BWC	1.27 (0.45)	1.87 (0.33)	1.12 (0.32)	1.79 (0.42)
Sexual Orientation	1.59 (0.90)	1.24 (0.55)	1.31 (0.70)	1.24 (0.55)
Sex life	1.36 (0.78)	1.22 (0.63)	1.24 (0.65)	1.39 (0.79)
First sexual experience	0.18 (0.39)	0.06 (0.24)	0.12 (0.32)	0.08 (0.35)

Note: Risk Eating Behavior (REB); Body Weight Concern (BWC).

**Table 8 healthcare-11-01682-t008:** Kruskal–Wallis for S.I. cause by independent variables.

Causes of SI	REB	Depression	BWC	Sexual Orientation	First Sexual Experience	Sexual Life
Out of curiosity	0.81	0.16	0.84	0.16	0.12	0.36
To try to feel better	**0.03**	**0.00**	0.16	0.79	0.14	0.96
To get my mind off my problems	**0.01**	**0.00**	0.21	0.49	0.40	0.41
To stop feeling or diminish my feelings of loneliness	**0.04**	**0.00**	0.64	0.30	0.58	0.26
To punish myself	**0.02**	**0.08**	0.55	0.69	0.13	0.15
To show others how angry I feel	0.45	0.93	0.72	0.17	0.42	**0.04**
To push away bad memories or sad things	0.15	0.80	0.92	0.98	0.91	0.79
To hurt important people in my life	0.28	0.16	0.12	0.56	0.55	0.75
To be different from others	0.14	0.33	0.44	0.31	0.67	0.49
To get attention	0.68	0.59	0.97	0.40	0.96	0.72
To show others how bad I feel	0.15	0.12	0.22	0.57	0.50	0.95

Note: Risk Eating Behavior (REB); Body Weight Concern (BWC). In bold, statistically significant data (*p* ≤ 0.05).

## Data Availability

The data presented in this study are available on request from the corresponding author. The data are not publicly available for ethical reasons.

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
