# Peer review of "Self-Injurious Behavior and Its Characteristics in a Sample of Mexican Adolescent Students"

_healthcare, 2023, doi:10.3390/healthcare11121682_

Round 1

Reviewer 1 Report

Self-injurious behavior and its characteristics in a sample of Mexican adolescent students

The article deals with self-injurious behaviors of youth in Mexico.

I am missing concrete results in the abstract. The results remain vague. There is no need to write that the results are compared with those of other researchers; this need not be mentioned. It would be better for the authors to draw a conclusion at the end of the summary as to what exactly we can have learn from their research.

The introduction reads well.

The participants were around two thirds women and one third men. Since self-injurious behaviors are more common among women, looking at all participants as a whole skews the data towards female participants.

Only students were examined, which significantly limits the generalizability of the data and represents a major disadvantage of the study.

The average age is 15-16 years, which is relatively young.

There is no comparison of the experimental and control groups with regard to the distribution of genders (are in both groups the same percentage of men? The same number of women?). The values given in Table 1 refers to the entire group. Overall, Table 1 should refer to the individual group in order to be able to compare whether important variables are the same in both groups. The table only gives values for the entire group.

Good description of the types of self-injurious behavior and frequency.

For the individual comparisons between the experimental and control groups, I lack the mean values plus/minus standard deviation (e.g. depression, sexual orientation, etc.).

A cardinal problem with this evaluation is that all participants who have already shown self-injurious behavior were thrown into one big pot. That is, a participant who once cut himself out of curiosity will be treated the same as a person who made deep cuts frequently and for years. A more intensive data analysis would be very desirable here.

Good and very detailed discussion of the data.

Reviewer 2 Report

Dear Authors, some suggestions:

The abstract is unclear. The purpose and conclusion of the study are not clearly understood from it. Also, in the abstract what does "OR" refer to?

One limitation of the study is that it is not randomized. This does not allow generalizing the results.

Furthermore, the results are consistent with what has already been highlighted by other studies carried out in Mexico. This is a strength but also a weakness.

I think it is necessary to elaborate the conclusions better to reinforce the appeal, the rationale and the conclusions that can be drawn on such a complex phenomenon.
